# Cellular Immune Profiling of Lung and Blood Compartments in Patients with SARS-CoV-2 Infection

**DOI:** 10.3390/pathogens12030442

**Published:** 2023-03-11

**Authors:** Letizia Santinelli, Alessandro Lazzaro, Francesca Sciarra, Luca Maddaloni, Federica Frasca, Matteo Fracella, Sonia Moretti, Alessandra Borsetti, Ginevra Bugani, Francesco Alessandri, Veronica Zullino, Franco Ruberto, Francesco Pugliese, Leonardo Sorrentino, Daniele Gianfrilli, Andrea Isidori, Mary Anna Venneri, Claudio M. Mastroianni, Giancarlo Ceccarelli, Gabriella d’Ettorre

**Affiliations:** 1Department of Public Health and Infectious Diseases, Sapienza University of Rome, 00185 Rome, Italy; 2Department of Experimental Medicine, Sapienza University of Rome, 00185 Rome, Italy; 3Laboratory of Virology, Department of Molecular Medicine, Affiliated to Istituto Pasteur Italia, Sapienza University, 00185 Rome, Italy; 4National HIV/AIDS Research Center, Italian Institute of Health, 00161 Rome, Italy; 5Department of General and Specialistic Surgery, Sapienza University of Rome, 00185 Rome, Italy; 6Azienda Ospedaliero Universitaria Policlinico Umberto I, 00185 Rome, Italy

**Keywords:** BALF, PBMC, SARS-CoV-2, cellular immune profile, severe COVID-19

## Abstract

Background: SARS-CoV-2 related immunopathology may be the driving cause underlying severe COVID-19. Through an immunophenotyping analysis on paired bronchoalveolar lavage fluid (BALF) and blood samples collected from mechanically ventilated patients with COVID-19-associated Acute Respiratory Distress Syndrome (CARDS), this study aimed to evaluate the cellular immune responses in survivors and non-survivors of COVID-19. Methods: A total of 36 paired clinical samples of bronchoalveolar lavage fluid (BALF) mononuclear cells (BALF-MC) and peripheral blood mononuclear cells (PBMC) were collected from 18 SARS-CoV-2-infected subjects admitted to the intensive care unit (ICU) of the Policlinico Umberto I, Sapienza University Hospital in Rome (Italy) for severe interstitial pneumonia. The frequencies of monocytes (total, classical, intermediate and non-classical) and Natural Killer (NK) cell subsets (total, CD56^bright^ and CD56^dim^), as well as CD4^+^ and CD8^+^ T cell subsets [naïve, central memory (TCM) and effector memory (TEM)], and those expressing CD38 and/or HLADR were evaluated by multiparametric flow cytometry. Results: Survivors with CARDS exhibited higher frequencies of classical monocytes in blood compared to non-survivors (*p* < 0.05), while no differences in the frequencies of the other monocytes, NK cell and T cell subsets were recorded between these two groups of patients (*p* > 0.05). The only exception was for peripheral naïve CD4^+^ T cells levels that were reduced in non-survivors (*p* = 0.04). An increase in the levels of CD56^bright^ (*p* = 0.012) and a decrease in CD56^dim^ (*p* = 0.002) NK cell frequencies was also observed in BALF-MC samples compared to PBMC in deceased COVID-19 patients. Total CD4^+^ and CD8^+^ T cell levels in the lung compartment were lower compared to blood (*p* = 0.002 and *p* < 0.01, respectively) among non-survivors. Moreover, CD38 and HLA-DR were differentially expressed by CD4^+^ and CD8^+^ T cell subsets in BALF-MC and in PBMC among SARS-CoV-2-infected patients who died from COVID-19 (*p* < 0.05). Conclusions: These results show that the immune cellular profile in blood and pulmonary compartments was similar in survivors and non-survivors of COVID-19. T lymphocyte levels were reduced, but resulted highly immune-activated in the lung compartment of patients who faced a fatal outcome.

## 1. Introduction

The Severe Acute Respiratory Syndrome Coronavirus 2 (SARS-CoV-2) is a ribovirus (a virus made of RNA) belonging to the β-CoV genus under the Coronaviridae family [1]. It is the etiological factor for the development of a clinically pleiotropic syndrome primarily involving the respiratory system and known by the name coronavirus disease 2019 (COVID-19). After its initial epidemic diffusion at the end of 2019, this coronavirus rapidly caused the COVID-19 pandemic, which is still ongoing, and whose epidemiologic course has dramatically changed from its start to now. Indeed, after the introduction of effective antiviral therapy and neutralizing monoclonal antibodies, together with different types of primary prophylactic strategies represented by several vaccination options against SARS-CoV-2 infection, from the beginning of 2021, an important drop in the infectivity and mortality of COVID-19 has been observed, which has continued to trend downwards until it reached an instable plateau at the end of 2022. Such instability seems to be whimsically balanced on the one hand by the virulence and pathogenicity of always newly stochastically selected emergent viral variants, and on the other by the selective pressure exerted by the therapeutical means available within the pharmaceutical inventory to suppress viral replication.

Although from a global health perspective, the COVID-19 pandemic has seen a reduction in alert degree because of the documented decrease in mortality, the COVID-19 disease itself has persistently represented a significant cause of morbidity and mortality throughout the world. Specifically, although most SARS-CoV-2-positive individuals have asymptomatic or mild-to-moderate diseases, a minority of those cases still develop severe pneumonia requiring mechanical ventilation, accounting for the high persistence of SARS-CoV-2-infection-associated morbidity and mortality [2,3]. Such data highlight how despite the large-scale distribution of effective pharmaceutical weapons against the viral spread of an initially mostly unknown enemy, the lack of deep knowledge of the virulence factor associated with SARS-COV-2, the interactions between the pathogen and the human host, and the pathogenic involvement of the innate and adaptive immune system represents the main obstacle to the proper management of this viral plague.

What is clear so far is that SARS-CoV-2-related systemically histologic damage can induce multi-organ dysfunction. This might aggravate pre-existing conditions, such as advanced age, obesity, diabetes mellitus, chronic obstructive pulmonary diseases, cerebro-cardiovascular chronic disease, chronic renal kidney disease, hepatic insufficiency, onco-hematologic active disease, primary and/or acquired immune deficiencies and neurodegenerative disease, which negatively influence the COVID-19 outcome, as it is expected in conditions that already engage the immune system [4]. Interestingly, it has been established that severe COVID-19 is triggered by a dysregulation in the early antiviral and adaptive immune responses with alterations in the interferon production, hyperinflammation and lymphopenia [5,6,7,8]. It is not surprising that the immune system appears to play a pivotal role in the complex interplay between pathogen and host, which result in a unique clinical presentation. As a matter of fact, critically ill people experiencing severe COVID-19 has been shown to exhibit a significant impairment of classical and loss of non-classical monocytes, as well as a reduced number and dysfunction of dendritic cells (DCs) and NK cells, followed by a decrease in CD4^+^ and CD8^+^ T cells that also become more activated [9,10,11]. However, to date, only a few studies have been performed on the cellular immune profile focusing on the lung and blood compartments of people with COVID-19, and specifically on those experiencing severe COVID-19 who underwent mechanical ventilation [12,13,14]. It is worth mentioning that the hematic and respiratory compartments are vastly different in terms of anatomical structure, histologic composition, cellular frequencies and immune response. Nonetheless, both systems are broadly involved during SARS-CoV-2 infection [15,16,17,18].

Approaching the big picture, a further clearer understanding of the relationships between the host cellular immune responses to SARS-CoV-2 infection and the distinct clinical presentations of COVID-19 represents a mandatory objective that the current research should address first and foremost, especially among those settings where this coronavirus infection presents its worst clinical outcomes, such as the experience of mechanical ventilation because of COVID-19-associated Acute Respiratory Distress Syndrome (CARDS). Unraveling the immunopathogenetic mechanisms underlying the progressive worsening of COVID-19 will help guide clinicians towards the selection of the best appropriate treatment strategy.

To provide further insight into the role played by the human host cellular immune response during SARS-CoV-2 infection within the context of severe CARDS, a comprehensive immunophenotyping analysis was carried out on paired bronchoalveolar lavage fluid (BALF-MC) and peripheral blood mononuclear cells (PBMC) samples collected from mechanically ventilated patients with (CARDS), evaluating the frequency of monocytes, NK and T cell subsets, as well as levels of T cell immune activation (CD38 and HLA-DR); moreover, the immune profiling of lung and blood compartments was performed in survivors and non-survivors of COVID-19.

## 2. Materials and Methods

### 2.1. Participants

A total of 36 paired clinical samples of BALF and peripheral blood were collected from 18 COVID-19 patients at the Division of ICU of Policlinico Umberto I, Sapienza University of Rome (Italy), during the first epidemic wave of SARS-CoV-2 (April–May 2020). Nasopharyngeal swabs for SARS-CoV-2 diagnosis, and BALF and whole blood for immunophenotyping analysis were collected at the time of ICU admission from patients coming directly from the emergency department; therefore, patients were not treated prior to sample collection. All patients were then treated after sample collection. RT-qPCR Detection of SARS-CoV-2 RNA nasopharyngeal swabs (RealStar SARS-CoV-2 RT-PCR, Altona Diagnostics) [19] and measurements of plasma C-reactive protein (CRP), D-dimer and albumin levels were performed for each patient analyzed. The study was approved by the institutional review board (Ethics Committee of Umberto I General Hospital, Rome), and all study participants gave written informed consent.

### 2.2. Specimen Processing and Cell Isolation

Bronchoalveolar lavage fluid specimens were collected in a sterile container after the instillation and recovery of sterile 0.9% normal saline solution during fiber-optic bronchoscopy in sedated patients, according to recommendations. Ten ml of filtered BALF was centrifuged for 10 min at 1800 rpm. The cell pellet was resuspended in FBS, supplemented with 10% DMSO and stored in liquid nitrogen until use.

Up to 20 mL of blood was collected from each SARS-CoV-2-infected patient in Vacutainer tubes containing ethylenediamine-tetraacetic acid (EDTA) (BD Biosciences, San Jose, CA, USA) and immediately processed. The isolation of PBMC was performed using Ficoll gradient centrifugation (Lympholyte; Cedarlane Labs, Hornby, ON, Canada), and cells were washed twice in PBS. PBMC were then stored in different aliquots in liquid nitrogen in Fetal Bovine Serum (FBS) supplemented with 10% dimethyl sulfoxide (DMSO) until use.

### 2.3. Immunophenotyping by Multiparametric Flow Cytometry assay

Thawed BALF and PBMC were washed twice with RPMI 1640 supplemented with 10% FBS, glutamine and antibiotics. The distribution of cellular phenotypes was evaluated on BALF and PBMC collected from SARS-CoV-2-positive patients by Beckman Coulter flow cytometer CytoFLEX-S. Along with side- and forward-scatter signals, signals were obtained from different fluorochrome-labeled anti-human monoclonal antibodies (mAbs): CD3-PerCP, CD4-APC-Vio770, CD8-FITC, CD14-APC, CD16-VioBlue, CD38-APC, HLA-DR-PE (Miltenyi Biotec, Bergisch Gladbach, Germany), CD4-APC-750 and CD56-AlexaFluor750 (BD Becton, Dickinson, San Jose, CA, USA). Monocytes were identified as CD3^−^CD14+ cells and were further divided into classical (CD14++CD16^−^), intermediate (CD14+CD16+) and nonclassical (CD14+CD16++) monocytes. NK cell subsets were identified as follows: total (CD14^−^CD19^−^CD3^−^CD56+), CD56^dim^ NK cells (CD56+CD16+) and CD56^bright^ NK cells (CD56++CD16^−^) [20]. T lymphocytes were identified as CD3+ cells after lymphocyte gating and were subsequently analyzed for surface expression of CD4 and CD8. CD4^+^ and CD8^+^ T cell subpopulations were identified according to the following phenotypic combinations: CD27+CD45RO^−^ naïve, CD27+CD45RO+ TCM and CD27^−^CD45RO+ TEM cells [21,22] (Appendix A). Cells were fixed before proceeding with cytofluorimetric analysis, considering the laboratory procedures for handling biological samples considered to be at risk. Thus, it was not possible to include a viability marker and stain the dead cells. Therefore, lymphocytes were gated on an SSC vs. FSC plot to exclude debris and FSC-H (Height) vs. FSC-A (area)/SSC-H vs. SSC-A plot to exclude doublets. A minimum of 10^5^ events were acquired for each sample. Gating strategies and data were analyzed using Software CytExpert (Beckman Coulter, Brea, CA, USA) and FlowJo V10 (TreeStar, Ashland, OR, USA) software.

### 2.4. Statistical Analysis

All data were analyzed with the Statistical Package for Social Science (SPSS) version 20, and all graphs were generated using GraphPad Prism (version 8.00). Baseline characteristics of SARS-CoV-2-positive participants were considered as mean ± standard deviation, median values and range (25°–75° percentile), simple frequencies (n) and proportions (percentages, %), according to the variable type, continuous or categorical, respectively, and the comparisons between survivors and non-survivors were performed using the “N-1” Chi-square test. The Mann–Whitney test was used to compare the frequencies of monocytes, NK cells and T lymphocyte subsets measured in BALF-MC and PBMC between survivors and non-survivors. The frequencies of monocytes, NK cells and T lymphocyte subsets measured among survivors and non-survivors were compared between BALF-MC and PBMC using paired comparisons by the Wilcoxon rank sum test. Statistical analyses were performed using GraphPad Prism software, version 9.4 (GraphPad Software Inc., La Jolla, CA, USA), and p values of less than 0.05 were considered statistically significant.

## 3. Results

### 3.1. Study Population

The present study included 18 SARS-CoV-2-positive participants (11 males and 7 females) with an average age of 72 (±10) years who were admitted to the ICU. At recruitment, all patients showed severe pneumonia confirmed by high-resolution chest tomography, had a median Pneumonia Severity Index (PSI) of 99 (84–110), and exhibited clinical manifestations of respiratory insufficiency that required invasive mechanical ventilation. The median length of the patients’ hospitalization was 54 (23–67) days from COVID-19 symptom onset; median CRP levels were 90,665 (37,925–155,070) mg/L, 2051 (1669–3922) mg/dL for D-dimer and 32 (29–34) mg/dL for albumin. COVID-19 patients admitted to the ICU suffered from lymphopenia, exhibiting a substantial decrease in T cell counts (median 665, range 420–765 cells/μL). During hospitalization, after sample collection, SARS-CoV-2-positive patients were treated as recommended at the time by the Italian Society of Infectious Diseases. Their median Charlson score [23] was 4 (2–5), while the median value of Comorbidity–Age–Lymphocyte count–Lactate dehydrogenase (CALL) score [24] was 11 (10–12).

A total of 12 of the 18 (67%) patients faced a fatal outcome. No significant differences were recorded in the clinical and demographic characteristics between surviving and non-surviving SARS-CoV-2-positive patients, with the only exception being D-dimer levels and days of hospitalization (Table 1).

### 3.2. Monocyte and NK Cell Subset Profiles in BALF-MC and PBMC of SARS-CoV-2-Infected Survivors and Non-Survivors

Given their pivotal function in host antiviral defense mechanisms and the potential danger posed by their dysregulation in severe SARS-CoV-2 infection [25,26], understanding monocytes and NK cell phenotypes is crucial for tackling the COVID-19 immunopathological mechanisms. Therefore, we evaluated the frequencies of monocytes and NK cell subsets in the peripheral and lung compartments of patients with and without a fatal COVID-19 outcome.

Surviving patients had higher frequencies of classical monocyte subsets at the peripheral level compared to non-survivors (Figure 1A, *p* < 0.01), while the percentages of total, non-classical and intermediate monocytes were similar in both groups of SARS-CoV-2-infected patients. Additionally, no differences in the respiratory frequencies of monocyte subsets were observed between survivors and non-survivors (Figure 1A, *p* > 0.05). Similarly, the evaluation of total, CD56^dim^ and CD56^bright^ NK cell frequencies found in BALF-MC and PBMC showed no differences between survivors and non-survivors (Figure 1B, *p* > 0.05).

### 3.3. T Cell Subset Profiles and Immune Activation in BALF-MC and PBMC of SARS-CoV-2-Infected Survivors and Non-Survivors

As the reduction in systemic levels of T lymphocytes might be considered among the predictors of mortality for patients with COVID-19 pneumonia [27,28], the frequencies and immune activation levels (CD38 and HLA-DR expression) of CD4^+^ and CD8^+^ T cell subsets were evaluated in BALF-MC and PBMC of survivors and non-survivors.

No differences in the total frequencies of CD4^+^ and CD8^+^ T cells and the CD4/CD8 ratio were found in peripheral and lung compartments between the two groups of SARS-CoV-2-infected patients (Figure 2A, *p* > 0.05).

However, levels of peripheral naïve CD4^+^ T cells were higher in survivors compared to deceased patients (Figure 2B, *p* = 0.04), while no differences in the frequencies of lung naïve CD4^+^ T cells were observed between these groups of patients (Figure 2B, *p* > 0.05). Similarly, CD4^+^ T cell (TCM and TEM) and CD8^+^ T cell subsets (naïve, TCM and TEM) frequencies evaluated in blood and lung compartments were not different between survivors and non-survivors (Figure 2B,C, *p* > 0.05). Regarding the evaluation of T cell immune activation levels in blood and lung samples, the expression of CD38 and HLA-DR markers by CD4^+^ and CD8^+^ T cell subsets resulted similar between survivors and non-survivors (*p* > 0.05; Figure 3A–F).

Having observed that patients with a fatal outcome of CARDS had reduced systemic frequencies of classical monocytes and naïve CD4^+^ T lymphocytes, we evaluated whether the frequencies of immune cells were different in the blood and lung compartments of SARS-CoV-2-infected patients.

As reported in Figure 1A, the frequencies of total, classical, non-classical and intermediate monocyte subsets were similar in the PBMC and BALF-MC of deceased patients. By contrast, these patients had a decrease in the levels of CD56^dim^ (*p* = 0.002) and increase in the levels of CD56^bright^ (*p* = 0.012) NK cell frequencies in BALF-MC samples compared to PBMC (Figure 1B). Additionally, the levels of total CD4^+^ and CD8^+^ T cells and those of CD4^+^ TCM cells observed in the lung compartment of non-survivors were also lower compared to those observed in the blood (*p* = 0.002, *p* = 0.012, and *p* = 0.012 respectively; Figure 2A,B).

The frequencies of CD4^+^ and CD8^+^ T cell subsets (naïve, TCM and TEM), single and combined, expressing CD38 and HLA-DR differed between the PBMC and BALF-MC of non-survivors (*p* < 0.05, Figure 3A–F); in particular, those who had a fatal COVID-19 outcome exhibited higher frequencies of CD4^+^ and CD8^+^ T cell subsets expressing single HLA-DR, or both CD38 and HLA-DR, in the lung compartment compared to peripheral blood (*p* < 0.05), except for naïve CD38+ and CD38+HLA-DR+ CD8^+^ T cells (*p* = 0.003 and *p* = 0.03, respectively), and CD38+ CD8^+^ TCM (*p* = 0.008) that were higher in PBMC compared to BALF-MC.

By contrast, the levels of CD38+HLADR+ CD4^+^ TCM (*p* = 0.03) and HLA-DR+ CD4^+^ TEM (*p* = 0.03), as well as of HLA-DR+ CD8^+^ TCM (*p* = 0.03) and CD38+HLA-DR+ CD8^+^ TEM (*p* = 0.04), were higher in the BALF-MC compared to the PBMC of survivors (*p* < 0.05, Figure 3A–F). Additionally, these patients had increased frequencies of naïve CD8^+^ T cells expressing CD38 (*p* = 0.03) and decreased levels of HLADR+ naïve CD8^+^ T cells (*p* = 0.03) in PBMC (Figure 3D) compared to the BALF-MC.

## 4. Discussion

Despite the remarkable advances against the SARS-CoV-2 infection, CARDS still represents a significant cause of admission to the ICU and death worldwide. In order to provide a clearer insight of the SARS-CoV-2 immune pathogenesis in the context of severe diseases such as CARDS, the aims of the current study were to profile the cellular immune response in both the circulatory and pulmonary compartments and to compare the immune responses of people experiencing a fatal outcome or surviving the infection.

With regard to the innate cellular immune system, monocytes/macrophages are deemed to be the chief promoters of the cytokine storm observed during SARS-CoV-2 infection, which has been identified as a potential pathogenic driving cause of COVID-19 patient conditions worsening and resulting in severe clinical outcomes, including CARDS and death [29]. Our evaluation of the impact of a SARS-CoV-2 infection on circulating and lung myeloid cell populations indicates an enrichment of classical monocytes in the blood of COVID-19 survivors. A decline in the frequencies of monocyte subsets in the blood was previously reported in people with SARS-CoV-2 infections; additionally, inflammatory transitional and non-classical monocytes appeared to be highly enriched in the lungs of individuals undergoing critical COVID-19 [30]. The expansion of activated myeloid cell subsets in the lungs might represent an in situ differentiation into macrophages from inflammatory non-classical monocytes, which may further contribute to COVID-19 progression [30]. However, our analysis showed that monocyte responses were similar in the blood and lungs of participants with SARS-CoV-2 infections who faced a fatal outcome, as well as among survivors. 

Evidence of immune involvement also points to the participation of NK cells in the lack of viral replication control, and to the regulation of adaptive responses during severe COVID-19 [31]. In this study, levels of total NK cells did not differ in the lungs and blood between survivors and non-survivors, while an increase in lung CD56^bright^ NK cells was found in patients with fatal outcomes, suggesting that the arming of CD56^bright^ NK cells with cytotoxic molecules correlates with a severe disease course [32]. Our analysis also showed a reduction in CD56^dim^ frequencies in the BALF-MC of deceased patients compared to PBMC, which may partially contribute to the failure of viral replication control in the lung compartment [33].

During severe SARS-CoV-2 infection, systemic levels of lymphocytes have been reported to undergo a dramatic drop as soon as the infection begins. Indeed, SARS-CoV-2 can directly promote an extreme stimulation and an exhaustive collapse of both CD4^+^ and CD8^+^ T cells [34,35,36]. To this extent, the literary data have reported that people with CARDS exhibit a drastic reduction in total T lymphocyte number [37], and SARS-CoV-2-infected patients admitted to the ICU have a reduction in circulating CD8^+^ T cells compared to non-ICU patients [38]. Prior reports on COVID-19 [12,39,40] showed lower frequencies of overall CD4^+^ and CD8^+^ T lymphocytes in BALF samples compared to the peripheral blood. Nonetheless, our results did not provide any robust evidence of different behavior in total T cell distributions according to CARDS outcome. However, the non-significant trend toward a difference in the frequencies of total CD4^+^ and CD8^+^ T cells in the blood and lung compartments between survivors and non-survivors observed in our analysis could be explained by our current sample size being too small to detect underlying statically significant differences.

It is known that viral infections, including that caused by SARS-CoV-2, can dysregulate the levels of several T cell subsets [36,41]. In agreement with this, our evaluation of the frequencies of naïve, central memory and effector memory CD4^+^ and CD8^+^ T cells showed that non-survivors exhibited lower levels of naïve CD4^+^ T lymphocytes in peripheral blood compared to survivors. These results support the hypothesis of an immune dysregulation in naïve CD4^+^ T cell homeostasis as a pivotal mechanism of CARDS pathogenesis. However, it has also been described that, in critical COVID-19 patients, CD4^+^ T cells may get stuck in a naïve state due to high levels of T-cells dysregulation [42], the observation of low frequencies of less differentiated CD4^+^ T cell subsets might reflect the continuous functional activation of T cells, leading to the accelerated consumption of naïve T cells through apoptosis or differentiation into specialized subsets, as already described for other viral infections [43]. Currently, it is unclear whether naïve T cells switch into effector/memory subsets or are depleted from both lung and peripheral blood compartments, but both mechanisms can likely occur [44,45,46]. 

As a matter of fact, the dysregulation of T cell frequencies does not necessarily entail their lower functionality; despite the fact that their immune activation levels do not seem to be associated with COVID-19 outcome, our flow cytometry analysis showed a higher T cell immune activation in BALF-MC than PBMC among non-survivors with CARDS. Considering that some data suggest that CD8^+^ T cells may have a hyperactivation signature in patients with severe COVID-19 [44,45,46], this might imply that strong CD38^+^HLA-DR^+^ or Ki67^+^ T cell activation could underline virus-specific CD8^+^ T cell responses in those patients [47,48]. Nevertheless, current data present potentially diversified patterns of CD4^+^ and CD8^+^T cell activation in patients with COVID-19 [48,49]. Despite a significant reduction in the frequency of T cells within the pulmonary compartment, these cells were strongly immune-activated in our series, as demonstrated by the expression of CD38 and/or HLA-DR markers by several CD4^+^ and CD8^+^ T cell subsets. These results might reflect different activation pathways within the hematic and respiratory compartments, with different enhanced functional responses at the site of infection for both helper and cytotoxic T cells involved in the immune response against SARS-CoV-2 [50,51].

In this regard, the anatomopathological series of both pulmonary autopsies and biopsies of people with SARS-CoV-2 infections at different stages of the infection have characterized the microscopic changes that occur during the progression of the pulmonary diseases, from local viral replication with minimal alveolo-endothelial injury to extensive vascular alterations and diffuse alveolar damage, which is the hallmark of ARDS. It is noteworthy that, despite the disease stage, an inflammatory lymphomonocytic infiltrate along the thickened interalveolar septa seems to be invariably present, which can degenerate into interstitial pneumonia [17]. Additionally, a significantly greater presence of T lymphocytes have been documented in COVID-19 lungs compared with controls [52], with a specific increase in CD3+ CD4^+^ T cells, particularly in the early phase of infection [18], outlining the quick and prominent T-lymphocyte response precisely at the site where viral replication is particularly active. Together with our findings, which documented lower hematic frequencies of naïve CD4 T cells and classical monocytes among non-survivors and a high immune-activation degree within the pulmonary site, these data suggest that in the context of CARDS-related lymphopenia, redistribution to the lung may be more clinically relevant for CD4^+^ T cells and classical monocytes, whose disfunction could contribute to or accelerate a fatal outcome, while the CD8^+^ T and NK cells might be more involved in the cytokine-triggered histological local damage, as reflected by the immune activation phenotype.

Despite these findings, our study had some limitations: this was an observational single-center study with a small sample size, which could cause some issues related to the frequency of immune cells in distinct anatomic systems of patients with severe COVID-19; moreover, to gain a better understanding of the immunopathological mechanisms, it would be advisable to analyze the levels of innate and adaptive immune cells together with cytokine patterns over time. We are also aware that the lack of a control group might fail to highlight the different immunological phenotypes of people infected with the SARS-CoV-2. An evaluation of the functional aspects of immune cells in both peripheral blood and lung compartments might provide a strong implementation for the results obtained in our study. To this extent, further studies need to be performed considering larger groups of patients, including healthy or non-COVID-19 (i.e., with other respiratory viral infections) individuals, to better focus on the local immune cellular patterns underlying the pathologic mechanisms associated with infection outcome. A deeper insight into the CARDS immunologic signature is crucial to improve the monitoring and treatment of pneumonia in people with severe COVID-19.

In conclusion, our findings pointed out that both peripheral blood and lung cellular responses play an important role against SARS-CoV-2 infection. Once again, we rediscovered how flow cytometry analysis provides considerable information for the immunological evaluation of people with SARS-CoV-2 infections, specifically among those experiencing severe CARDS. The current work highlights the relevance of both bronchoalveolar lavage and peripheral blood cells as surrogate markers to study the cellular immunity in the course of complex infectious diseases such as COVID-19.

## Figures and Tables

**Figure 1 pathogens-12-00442-f001:**
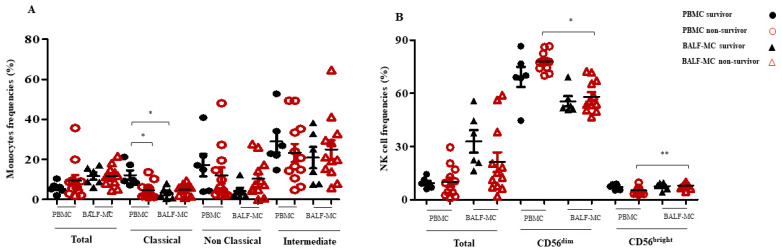
Frequency of monocytes (total, classical, intermediate and non-classical) and NK cells (dim and bright) in lung and blood compartments of COVID-19 survivors and non-survivors. The frequencies of total, classical, intermediate and non-classical monocytes (**A**) and of total, CD56dim and CD56bright NK cell subsets (**B**) were measured in BALF-MC and PBMC of COVID-19 survivors and non-survivors. *p* values < 0.05 were considered statistically significant (* significant, ** highly significant).

**Figure 2 pathogens-12-00442-f002:**
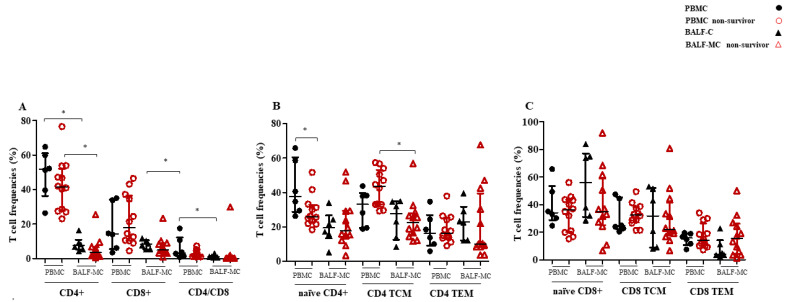
CD4^+^ and CD8^+^ T cell subset frequencies (total, naïve, TCM and TEM) and CD4/CD8 ratio in lung and blood compartments of COVID-19 survivors and non-survivors. The frequencies of CD4^+^ and CD8^+^ T cells and CD4/CD8 ratio (**A**), and of CD4^+^ and CD8^+^ naïve, TCM and TEM subsets (**B**,**C**) were measured in BALF-MC and PBMC of COVID-19 survivors and non-survivors. *p* values < 0.05 were considered as statistically significant (* significant).

**Figure 3 pathogens-12-00442-f003:**
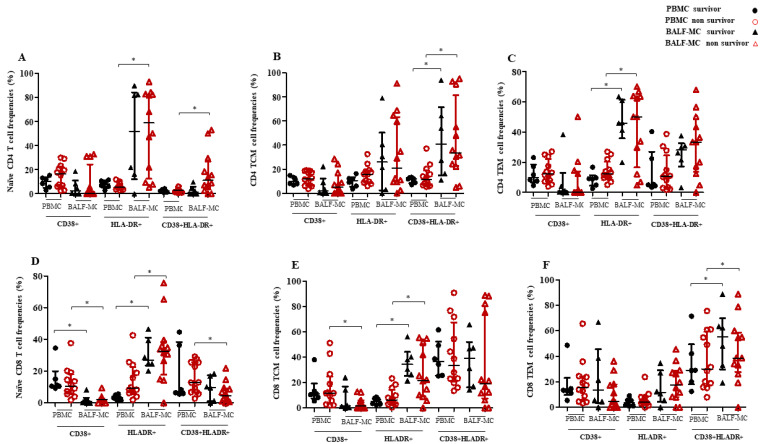
CD4^+^ and CD8^+^ T cell subset frequencies (naïve, TCM, TEM) expressing immune activation markers in lung and blood compartments of COVID-19 survivors and non-survivors. The frequencies of CD4^+^ (**A**–**C**) and CD8^+^ (**D**–**F**) naïve, TCM and TEM expressing CD38 and HLA-DR markers (single and both), were measured in BALF-MC and PBMC of COVID-19 survivors and non survivors. *p* values < 0.05 were considered as statistically significant (* significant).

**Table 1 pathogens-12-00442-t001:** Demographical and clinical characteristics of surviving and non-surviving patients with CARDS.

Item ^a^	Survivors (n = 6)	Non-Survivors (n = 12)	*p* Value ^b^
Gender Male/Female [number (%)]	[4 (66.7)]/[2(33.3)]	[7 (62.5)]/[5(37.5)]	0.86/0.86
Age (years)	71 (±16)	72 (±7)	
White blood cells (cells/mm^3^)	8735 (6875–10,115)	9200 (4545–14,410)	0.925
Neutrophils (cells/mm^3^)	7260 (5717–8525)	7775 (3702–12,930)	1.00
Lymphocytes (cells/mm^3^)	805 (535–1007)	600 (417–750)	0.223
Monocytes (cells/mm^3^)	235 (222–435)	290 (267–577)	0.301
C-reactive protein (mg/L) (cells/mm^3^)	68,100 (37,925–104,125)	90,765 (49,365–182,720)	0.453
D-dimer (mg/dL)	4473 (4404–4473)	1949 (1576–4109)	0.034
Albumin (mg/dL)	34 (31–34)	30 (29–32)	0.115
Lactate dehydrogenase (U/L)	423 (305–452)	500 (350–533)	0.134
Platelets (cells/mm^3^)	220,500 (175,250–226,750)	264,500 (201,500–332,000)	0.160
Length of hospitalization (days)	101 (66–130)	33 (12–49)	0.002
CHARLSON index	4 (2–5)	4 (2–5)	0.658
CURB-65	2 (1–2)	2 (2–2)	0.653
EXP CURB-65	4 (3.5–4.5)	4 (4–4)	1.00
PSI	110 (79–112)	96 (87–100)	0.446
CALL score	10 (9–12)	11 (10–13)	0.224

^a^ Data were expressed as median (range) or numbers (percentages); ^b^
*p* values < 0.05 were considered statistically significant. Abbreviations: LDH: lactate dehydrogenase; EXP CURB-65: Expanded CURB-65; PSI: Pneumonia Severity Index; CALL score: Comorbidity–Age–Lymphocyte count–Lactate dehydrogenase (CALL) score.

## Data Availability

The datasets analyzed for this study can be provided by the corresponding author upon reasonable request.

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
