# Peer review of "Cellular Immune Profiling of Lung and Blood Compartments in Patients with SARS-CoV-2 Infection"

_pathogens, 2023, doi:10.3390/pathogens12030442_

Round 1

Reviewer 1 Report

In the article "Cellular Immune profiling of lung and blood compartments in patients with SAR-Co V-2 infection", the authors Santinelli and Lazzaro et al., have demonstrated the profile of the immune cells in the PBMC and bronchoalveolar lavage fluid (BALF). The study is interesting but at the same time, it requires a few pieces of information to be incorporated into the current manuscript, which might help the readers to understand the data well. A list of such suggestions is listed below:

1. Incorporation of a figure discussing the gating strategy of the flow cytometry data.
2. Which kind of dye was used to stain the dead cells to gate them out of the analyzed population

3. Did any of the patients undergo any treatment before the samples were collected? Drugs can possibly affect the immune profile of the patients.

4. All the figures present in the manuscript would be easier to comprehend when shown in different colors or patterns.

5. As a control, a side-by-side comparison with the immune profile of the PBMC and BALF specimens from a cohort of healthy people would directly impact the different immunological phenotypes of people infected with the SARS-CoV-2 virus.

Author Response

Reviewer 1 

Re: We thank the Reviewer for his/her comments and helpful suggestions. Below is a point-by-point response to his/her criticisms. The manuscript has been revised accordingly and we hope it now meets the Reviewer’s demands. For the Reviewer’s convenience, the revisions have been highlighted in the manuscript. 

  • Incorporation of a figure discussing the gating strategy of the flow cytometry data.  

We thank Reviewer#1 for the constructive criticism for improving the quality of our manuscript. We provided gating strategies of our immunophenotyping analysis as a supplementary file (Figure S1). 

  • Which kind of dye was used to stain the dead cells to gate them out of the analyzed population 

We thank the reviewer for this helpful comment that will allow us to better explain our results.  

As reported, we collected this samples during the first pandemic wave, when Italian health authorities had not yet provided any indication regarding the safety of the samples collected towards SARS-CoV-2 transmission. Therefore, the cells were fixed before proceeding with cytofluorimetric analysis, considering the laboratory procedures for handling biological samples considered to be at risk. Thus, it was not possible to include the viability marker and stain the dead cells. However, we gate lymphocytes on SSC vs FSC plot to exclude debris and FSC-H (Height) vs. FSC-A (area)/SSC-H vs SSC-A plot to exclude doublets.  

  • Did any of the patients undergo any treatment before the samples were collected? Drugs can possibly affect the immune profile of the patients. 

Thanks for this observation. All samples were collected at the time of ICU admission from patients directly coming from the emergency department; therefore, patients were not treated prior to sample collection; we clarified this issue in the material and methods section. Anyway, all patients were then treated with the best available therapy, as reported in the results section. 

  • All the figures present in the manuscript would be easier to comprehend when shown in different colors or patterns. 

We modified pattern and colors in all figures, as suggested. 

  • As a control, a side-by-side comparison with the immune profile of the PBMC and BALF specimens from a cohort of healthy people would directly impact the different immunological phenotypes of people infected with the SARS-CoV-2 virus. 

We thank the Reviewer#1 for this observation, and we absolutely agree with him/her. We are aware that limitations of our analysis include the lack of a control group (likewise previous studies from literature with similar outcome, Dentone, et al, BMC infectious diseases 2021), which might affect the power of the study to detect differences and associations between enrolled participants. It is worthy of mentioning that although the collection of BALF is safe, it still represents an invasive procedure; thus, it remains difficult and unethical to perform BALF sampling in a group of healthy people. In addition, the study protocol approved by the ethics committee did not include the enrollment of a control group, also considering that we compared two groups of patients with different COVID-19 outcome.  We discussed this issue as a study limitation. 

Reviewer 2 Report

Authors did extensive cellular profiling of the survivor and Non Survivors with SARS-CoV-2 infection. 

Minor comments:

Did authors performed  functional aspects of Immune cells in peripheral blood and lungs? Will be huge contribution in the field.

Authors can provide gating strategies of the proposed Immunophenotyping in supplementary figures. I guess that will give more visibility to the novel findings.

Author Response

Reviewer 2 

Re: We thank the Reviewer for the helpful suggestions to improve our work. The manuscript has been revised accordingly and we hope it now meets the Reviewer’s demands. For the Reviewer’s convenience, the revisions have been highlighted in the manuscript. 

  • Did authors performed functional aspects of Immune cells in peripheral blood and lungs? Will be huge contribution in the field.  

We thank Reviewer#2 for the opportunity to improve the quality of our manuscript according to his/her suggestions. We absolutely agree that functional analysis provide a strong contribution to the results obtained in our study, but unfortunately, we were not able to collect enough sample to perform functional evaluation, especially from bronchoalveolar lavage fluid specimens. We will provide this information as a study limitation. Nevertheless, we are highly confident that the results obtained in this study remain an interesting source of data. 

  • Authors can provide gating strategies of the proposed immunophenotyping in supplementary figures. I guess that will give more visibility to the novel findings.  

As suggested, we provided gating strategies of our immunophenotyping analysis as a supplementary file. 

Round 2

Reviewer 1 Report

I would like to thank the author for clarifying all the concerns. The manuscript seems easier to comprehend and has the potential to be published in its current form.

Author Response

We appreciate the Reviewer’s positive feedback and we thank the Reviewer for his/her positive assessment of our manuscript.